# COVID-19 and mental health deterioration by ethnicity and gender in the UK

**Eugenio Proto**[1,2,3☯], **Climent Quintana-Domeque**[ID][3,4,5☯]

**1** Department of Economics, Adam Smith Business School, University of Glasgow, Glasgow, United Kingdom, **2** CEPR, London, United Kingdom, **3** IZA, Bonn, Germany, **4** Department of Economics, Business School, University of Exeter, Exeter, United Kingdom, **5** Department of Economics, HCEO, University of Chicago, Chicago, IL, United States of America

☯ These authors contributed equally to this work.
* c.quintana-domeque@exeter.ac.uk

## Abstract

We use the UK Household Longitudinal Study and compare pre-COVID-19 pandemic (2017-2019) and during-COVID-19 pandemic data (April 2020) for the same group of individuals to assess and quantify changes in mental health as measured by changes in the GHQ-12 (General Health Questionnaire), among ethnic groups in the UK. We confirm the previously documented average deterioration in mental health for the whole sample of individuals interviewed before and during the COVID-19 pandemic. In addition, we find that the average increase in mental distress varies by ethnicity and gender. Both women –regardless of their ethnicity– and Black, Asian, and minority ethnic (BAME) men experienced a higher average increase in mental distress than White British men, so that the gender gap in mental health increases only among White British individuals. These ethnic-gender specific changes in mental health persist after controlling for demographic and socioeconomic characteristics. Finally, we find some evidence that, among men, Bangladeshi, Indian and Pakistani individuals have experienced the highest average increase in mental distress with respect to White British men.

## Introduction

In this paper we investigate the impact of a massive negative health and economic shock, the COVID-19 pandemic, on mental wellbeing by ethnicity and gender in the UK. This is an interesting outcome for economists and policymakers alike, if only because of the well-established link between psychological wellbeing and productivity [1]. We compare the average mental wellbeing –as measured by the General Health Questionnaire (GHQ-12) [2]– of the same group of individuals interviewed before the pandemic (2017-2019) and early in the pandemic (April 2020).

In the UK, 1 out of 4 deaths in March and April 2020 (38,156 deaths) involved the Coronavirus [3], there were 177,487 cumulative positive cases until April 30 [4], and the level of economic activity in the UK plummeted by 15.7 points between the first quarter and the second quarter of 2020 [5]. Moreover, these health and economic costs are affecting

**Data Availability Statement:** Understanding Society data are available through the UK Data Service. Researchers who would like to use Understanding Society need to register with the UK

Data Service before being allowed to apply for or download datasets. More information: https://www.understandingsociety.ac.uk/documentation/access-data.

**Funding:** The authors received no specific funding for this work.

**Competing interests:** The authors have declared that no competing interests exist.

disproportionately some groups more than others. Ethnic minority individuals have a substantially higher risk of COVID-related death than white people [6–10], and the COVID-induced economic contraction and shutdown can especially impact minority ethnic individuals [9, 11, 12].

This differential mortality risk by ethnicity can be driven by a higher risk of acquiring infection (e.g., if ethnic minority individuals are more likely to be employed as "key workers", which are subject to a higher risk of infection), a higher risk of poor outcomes once infected (e.g., if ethnic minority individuals are more likely to suffer from underlying health conditions), or both [13]. For instance, the Indian ethnic population represents 14% of doctors in England and Wales, but only 3% of the working-age population [12], and recent reports show that the average Black, African and Ethnic Minority (BAME) risk of infection is 56% higher than the White British risk for working-age people, and 69% higher for those 65 plus [14]. Bangladeshis are more than 60% more likely to have a long-term health condition compared with White British aged 60 plus [12]. However, only a small part of the excess COVID-19 mortality risk of ethnic minority groups can be explained by comorbidities, deprivations, or other factors [10].

The differential economic impact by ethnicity can be driven by unemployment, income loss, or financial insecurity. Pakistani men are 70% more likely to be self-employed than the White British majority [12], and the incomes of self-employed workers are more uncertain. In addition, men from minority ethnic groups are more likely to be affected by the shutdown [12]: Bangladeshi men are four times as likely as White British men to have jobs in shut-down industries (e.g. restaurant sector), and Pakistani men are nearly three times as likely as White British men (e.g. taxi driving sector).

The COVID-19 pandemic can be considered a traumatic event [15], which may lead to mental health deterioration for multiple reasons [15–17]. In the UK, there has been an increase in mental distress between the pre-pandemic and pandemic periods, stronger among women and younger individuals [18]. Given the differences in mortality risk and financial security across different ethnic populations, we may expect differential effects on mental health too. Indeed, in the US, racial/ethnic minorities reported having experienced disproportionately worse mental health outcomes [19]. While in the UK there were no differential changes in mental health problems between White and non-White individuals from 2017-2019 to April 2020 [16], this comparison may mask important differences across ethnic populations [12, 16]. Moreover, while a lot has been documented on gender inequality and the pandemic [18, 20–22], the potential interaction between gender and ethnicity requires further investigation, if only because of the different "gender roles" within households across different ethnic populations. For instance, 29% of Bangladeshi working-age men both work in a shut-down sector and have a partner who is not in paid work compared with only 1% of White British men [12].

We use data from the UK Household Longitudinal Study (UKHLS) to complement and extend upon previous research by documenting a decline in mental health between before and during the COVID-19 pandemic by ethnicity and gender in the UK. We quantify the average change in the GHQ-12 score (described in the Materials and methods section) from 2017-2019 to April 2020 between different population groups (defined by ethnicity and gender), and describe systematic differences in mental health deterioration due to the COVID-19 pandemic.

We conduct two types of analyses. First, we present a graphical analysis which displays raw (unadjusted) differences in the change in mental health from 2017-2019 to April 2020 by ethnicity and gender. Second, we run a regression analysis which compares unadjusted vs. adjusted changes. The purpose of this analysis is to show whether the ethnic-gender specific changes in mental health can be explained by differences in demographic or socioeconomic

variables that can act a mediators or be affected by ethnicity and gender characteristics. Thus, our investigation does not make causal claims. While we do not model causal chains [13], we provide a first approximation to the impact of the COVID-19 pandemic on mental health by ethnicity and gender in the UK.

We confirm the previously documented average deterioration in mental health for the whole sample of individuals interviewed before and during the COVID-19 pandemic. In addition, we find that the increase in average mental distress varies by ethnicity and gender. First, BAME men experience a higher average increase in mental distress than White British men. Second, women –regardless of ethnicity– experience a higher average increase in mental distress than White British men. A by-product of these two findings is that the gender gap in mental health increases only among White British individuals. These findings are robust to controlling for existing differences in demographic and socioeconomic characteristics. Finally, we find some evidence that, among men, Bangladeshi, Indian and Pakistani (BIP) have experienced the highest average increase in mental distress with respect to White British men.

We believe ours is an important, albeit preliminary, step in identifying how to address the unequal impact of the pandemic on mental health inequality. While our analysis does not shed light on the actual mechanisms underlying the predictive power of ethnicity in explaining the increase in mental distress, the similarity of both raw (unadjusted) and adjusted changes allows us to rule out differences in demographic and socioeconomic characteristics as the drivers of the differential change in mental health from 2017-2019 to April 2020 by ethnicity and gender.

## Materials and methods

### Data

We use two waves of data from the UK Household Longitudinal Study (UKHLS or Understanding Society) [23, 24], wave 9 (2017-2019) and the first monthly COVID-19 wave (April 2020) and join others [16, 18, 20, 21, 25, 26] in their effort to understand the effects of the COVID-19 pandemic and the lockdown on mental wellbeing. We use sampling weights [27, 28].

We combine the two waves to generate a dataset of 53,816 observations with two components, one cross-sectional, the other longitudinal, and drop 4,660 observations with missing information on the variable used to define our three measures of mental health, the 12-item General Health Questionnaire (GHQ-12, described in the next subsection). Table 1 reports how many observations with information on the GHQ-12 belong to each wave and how many individuals are observed in both waves. Among individuals interviewed in 2017-19 with information on the GHQ-12, 45.8% were re-interviewed in April 2020 (unweighted: 43.8%, 14,523/ 33,143).

While the attrition rate is substantial, more than 50%, it is consistent with previous research using the same data [16, 27]. More importantly, selective attrition based on the level of mental

**Table 1. Cross-sectional and longitudinal dimensions.**

|  | Individual is observed twice | | |
|---|---|---|---|
| **Wave** | **No** | **Yes** | **Total** |
| 2017-2019 | 18,620 | 14,523 | 33,143 |
| April 2020 | 1,490 | 14,523 | 16,013 |
| Total | 20,110 | 29,046 | 49,156 |

Authors' elaboration using UKHLS data: wave 9 (2017-2019) and April 2020 COVID-19 wave.

distress at baseline is negligible. A one standard deviation increase in the GHQ-12 in 2017-2019 translates into an increase of 1.28% in the likelihood of attrition [95% CI: 0.55, 2.0], or 2.8% of the attrition rate, as estimated by means of an OLS regression using weights and robust standard errors but no controls. This is reassuring, and consistent with previous research reporting that mental health problems in 2017-2019 are not related to participation in the COVID-19 wave [16].

## The 12-item General Health Questionnaire

In order to identify changes in mental wellbeing from 2017-2019 and April 2020, we use three measures based on the 12-item General Health Questionnaire, GHQ-12 [2]. The GHQ-12 is a well-known self-report instrument for evaluating mental health where the respondent must report the extent to which 12 symptoms are present in the past few weeks on a Likert scale (see S1 Appendix), and its range goes from 0 to 36. The first measure we use is the difference in the GHQ-12 from 2017-2019 to April 2020. The second measure is the standardised change in the GHQ-12 between waves, so that differences in changes in the GHQ-12 among ethnic groups are measured in standard deviations. Finally, we use a third measure based on the change of a binary indicator of being at risk of presenting with mental health problems (GHQ "caseness" score) [16], which has been validated against psychiatric interviews [29, 30]. If individuals report experiencing at least 3 of the 12 symptoms, we classify them at risk of mental health problems (see S1 Appendix).

## Regression analysis

We perform two types of regressions: short regressions, as in (1), and long regressions, as in (2). We use OLS to estimate three types of short regressions:

$$\Delta y_i = \alpha^S + \beta^S Female_i + \gamma^S BAME_i + \delta^S Female_i \times BAME_i + e_i^S, \qquad (1)$$

where $\Delta y_i$ is either (i) the change (-36, 36) in the GHQ-12 score, (ii) the standardised change in the GHQ-12 score, or (iii) the difference in the GHQ "caseness" score (-1,1) from 2017-2019 to April 2020 for individual $i$; $Female_i$ = 1 if individual $i$ is a woman, 0 if the individual is a man; $BAME_i$ = 1 if individual $i$ belongs to the BAME population, 0 if the individual belongs to the White British population; and $e_i^S$ is a regression residual. $\beta^S$ captures the average difference between women and men in the change in GHQ-12 between 2017-2019 and April 2020 among White British individuals; $\gamma^S$ captures the average difference between BAME and White British individuals in the change in GHQ-12 between 2017-2019 and April 2020 among men; finally, $\gamma^S + \delta^S$ captures the average difference between BAME and White British individuals in the change in GHQ-12 between 2017-2019 and April 2020 among women.

We also use OLS to estimate three types of long regressions:

$$\Delta y_i = \alpha^L + \beta^L Female_i + \gamma^L BAME_i + \delta^L Female_i \times BAME_i + \pi \mathbf{X}_i + e_i^L, \qquad (2)$$

where $\mathbf{X}_i$ is a vector of other demographic and socioeconomic variables, including age group indicators, month of interview indicators in 2017-2019, a face-to-face interview indicator in 2017-2019 [31], household size in April 2020, an indicator for living in a couple in April 2020, place of residence (location) indicators, qualification indicators in 2017-2019, employment status indicators in 2017-2019, net personal income in 2017-2019, and an indicator of having at least one health condition in April 2020 (all variables are defined in detail in S2 Appendix). $\beta^L$ captures the adjusted average difference between women and men in the change in GHQ-12 between 2017-2019 and April 2020 among White British individuals; $\gamma^L$ captures the adjusted

average difference between BAME and White British individuals in the change in GHQ-12 between 2017-2019 and April 2020 among men; finally, $\gamma^L + \delta^L$ captures the adjusted average difference between BAME and White British individuals in the change in GHQ-12 between 2017-2019 and April 2020 among women.

All regressions are estimated using weights and robust standard errors. We then repeat our analysis replacing the indicator BAME with two indicators, BIP (Bangladeshi, Indian and Pakistani) and non-BIP (White Other, Mixed, Black, Asian, and Arab), and the interaction Female and BAME with two interactions, Female and BIP, and Female and non-BIP.

## Results

### Descriptive statistics

Pooling the two cross-sections of data, we find that the average mental distress (GHQ-12: 0-36) has increased from 11.44 [95% CI: 11.36, 11.52] in 2017-2019 to 12.52 [95% CI: 12.40, 12.65] in April 2020, a 0.19 standard deviation increase [95% CI: 0.17, 0.21]. Exploiting the longitudinal dimension of the dataset (i.e., comparing the same individuals before and after), we document a similar change, from 11.28 [95% CI: 11.17, 11.40] to 12.51 [95% CI: 12.38, 12.63], a 0.21 standard deviation increase [95% CI: 0.19, 0.23]. In what follows, we focus on the longitudinal dimension of the dataset.

Table 2 contains a description of our sample of (panel) individuals regarding ethnicity and gender. 91.5% of the individuals are White British and the remaining 8.5% are BAME. The 8.5% of BAME is the sum of approximately 2% Bangladeshi, Indian and Pakistani (BIP), and 6.5% of other minority ethnic groups (non-BIP: White Other, Mixed, Black, Asian, and Arab). 55.5% of our sample participants are women. By gender, 91.3% of women are White British and the remaining 8.7% are BAME (1.8% are BIP and 6.9% are non-BIP), and 91.8% of men are White British and the remaining 8.2% are BAME (2.1% are BIP and 6.1% are non-BIP).

**Table 2. Description of the sample of individuals observed in 2017-2019 and April 2020.**

|  | N | % |
|---|---|---|
| **Ethnicity** |  |  |
| White British | 11,451 | 91.5 |
| BAME | 1,066 | 8.5 |
| BIP | 247 | 2.0 |
| Non-BIP | 819 | 6.5 |
| **Gender** |  |  |
| Female | 7,012 | 55.5 |
| Male | 5,615 | 44.5 |
| **Ethnicity and gender** |  |  |
| White British Female | 6,628 | 91.3 |
| BAME Female | 635 | 8.7 |
| BIP Female | 134 | 1.8 |
| Non-BIP Female | 501 | 6.9 |
| White British Male | 4,821 | 91.8 |
| BAME Male | 432 | 8.2 |
| BIP Male | 112 | 2.1 |
| Non-BIP Male | 320 | 6.1 |

Authors' elaboration using UKHLS data: wave 9 (2017-2019) and April 2020 COVID-19 wave. Statistics (N and %) are weighted using the survey sample weights (see S2 Appendix).

## BAME vs. White British

Fig 1 displays the average mental distress (on a scale from 0-36) in 2017-2019 and April 2020 by ethnicity (BAME vs. White British) and gender (women vs. men). Women report a higher average level of mental distress than men do –within each ethnic group, there is a gender gap in mental distress in both periods– and all groups experience an average increase in mental distress from 2017-2019 to April 2020. Interestingly, however, the increase in mental distress varies by ethnicity and gender.

Indeed, Fig 2 plots the average change in GHQ-12 between the two periods, and shows that both women –regardless of their ethnicity– and BAME men are the groups experiencing higher changes in mental distress: 1.7 units (BAME women), 1.6 units (White British women), 1.5 units (BAME men) and 0.6 units (White British men). Given that the average change in mental health among women does not vary by ethnicity, the gender gap in mental health increases only among White British individuals.

The patterns documented in Figs 1 and 2 do not account for demographic and socioeconomic differences across groups. Table 3 displays the average of some demographic and socioeconomic variables by ethnicity within the groups of men (panel A) and women (panel B). Among the individuals in our sample, BAME individuals tend to be younger than White British individuals: the average gap is 7.3 years among men (p-value = 0.000) and 6 among women (p-value = 0.000). We also observe statistically significant differences in household size: BAME men tend to live in larger households (average gap of about 0.6 members, p-value = 0.000) than White British men, and similarly BAME women tend to live in larger households than White British women, although the average gap is halved (about 0.3 members, p-value = 0.000). There

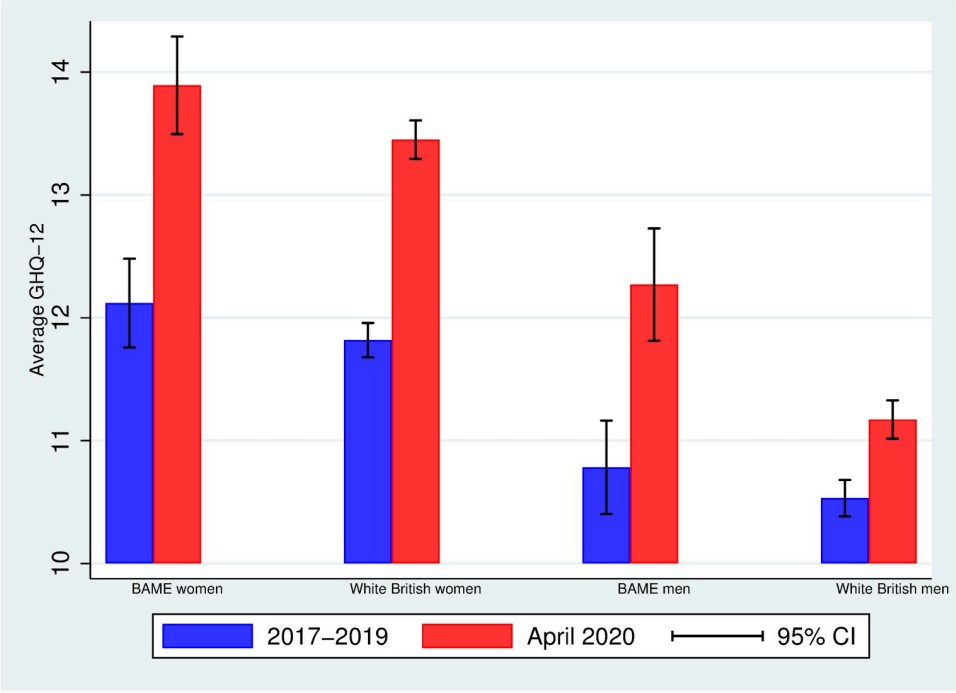

**Fig 1. Average GHQ-12 in 2017-2019 and April 2020 among BAME and White British individuals by gender.**
Authors' elaboration using UKHLS data: wave 9 (2017-2019) and April 2020 COVID-19 wave. Observations are weighted using the survey sample weights (see S2 Appendix). 95% CI: 95% approximate confidence intervals (point estimate ± 1.96 times the standard error of the point estimate).

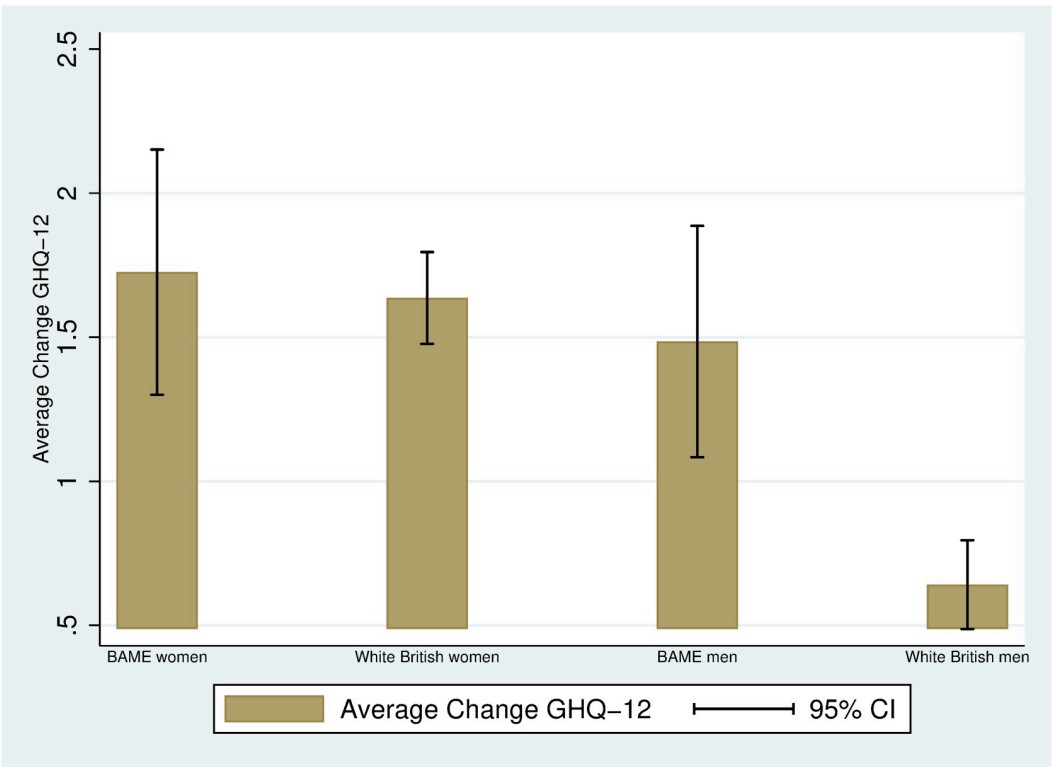

**Fig 2. Average change in GHQ-12 between 2017-2019 and April 2020 among BAME and White British individuals by gender.** Authors' elaboration using UKHLS data: wave 9 (2017-2019) and April 2020 COVID-19 wave. Observations are weighted using the survey sample weights (see S2 Appendix). 95% CI: 95% approximate confidence intervals (point estimate ± 1.96 times the standard error of the point estimate).

are also large differences in location: 31-33% of BAME individuals live in the London area (p-value = 0.000), while the percentage among White British is 8-9% (p-value = 0.000). BAME individuals are 15-18 percentage points more likely to hold a BA (or higher) than White British individuals. We also find differences in family care or home by ethnicity among women: 8.5% of BAME women report family care or home as their employment situation, while this is 4.6% among White British women. No average differences in personal monthly income are observed, although BAME individuals tend to be more qualified than White British individuals. Finally, BAME individuals are less likely to report having a health condition (8 percentage points less likely among men, 11% less likely among women), which is consistent with the fact that they are also younger.

It is important to understand whether the ethnic-gender specific changes in mental distress from 2017-2019 to April 2020 persist after controlling for the existing demographic and socio-economic differences across groups reported in Table 3. Table 4 contains the short and long regressions (see Materials and methods) for the three measures of mental health deterioration: the difference in the GHQ-12 score (as in Figs 1 and 2), the standardised difference, and the difference in the GHQ "caseness" score. Column (1) replicates the graphical findings in Fig 2: first, among White British individuals, between 2017-2019 and April 2020, women have experienced a higher increase –1 unit more (SE = 0.13)– in mental distress than men; second, among men, between 2017-2019 and April 2020, BAME individuals have experienced a higher increase –0.84 units more (SE = 0.31)– in mental distress than men; finally, among women,

**Table 3. Average demographic and socioeconomic characteristics: BAME and White British.**

| | Age | Household size | London | BA or higher | Self-employed | Family care | Personal income | Health conditions |
|---|---|---|---|---|---|---|---|---|
| Panel A. Men | | | | | | | | |
| White British | 54.56 | 1.77 | 0.084 | 0.356 | 0.105 | 0.005 | 2.17 | 0.521 |
| | N = 4,541 | N = 4,541 | N = 4,539 | N = 3,991 | N = 4,525 | N = 4,525 | N = 4,541 | N = 4,541 |
| BAME | 47.29 | 2.33 | 0.327 | 0.535 | 0.125 | 0.0005 | 2.18 | 0.442 |
| | N = 712 | N = 712 | N = 711 | N = 623 | N = 709 | N = 709 | N = 712 | N = 712 |
| BAME – White British | -7.27*** | 0.56*** | 0.243*** | 0.179*** | 0.020 | -0.0045*** | 0.010 | -0.079*** |
| | [0.000] | [0.000] | [0.000] | [0.000] | [0.270] | [0.000] | [0.897] | [0.004] |
| Panel B. Women | | | | | | | | |
| White British | 51.76 | 1.83 | 0.087 | 0.317 | 0.064 | 0.046 | 1.45 | 0.514 |
| | N = 6,196 | N = 6,196 | N = 6,195 | N = 5,440 | N = 6,143 | N = 6,143 | N = 6,196 | N = 6,196 |
| BAME | 45.77 | 2.11 | 0.314 | 0.464 | 0.097 | 0.085 | 1.55 | 0.407 |
| | N = 1,067 | N = 1,067 | N = 1,067 | N = 943 | N = 1,060 | N = 1,060 | N = 1,067 | N = 1,067 |
| BAME – White British | -5.99*** | 0.28*** | 0.227*** | 0.147*** | 0.033** | 0.039*** | 0.10 | -0.107*** |
| | [0.000] | [0.000] | [0.000] | [0.000] | [0.027] | [0.001] | [0.134] | [0.000] |

Authors' elaboration using UKHLS data: wave 9 (2017-2019) and April 2020 COVID-19 wave. Age (years), household size (number of people), London (0-1), BA or higher (0-1), Self-employed (0-1), Family care (0-1), Personal income (in £1,000), and Health conditions (0-1). Statistics are weighted using the survey sample weights (see S1 and S2 Appendices). p-values in brackets.

* $p < 0.10$,

** $p < 0.05$,

*** $p < 0.01$.

there is no evidence of a differential increase in mental distress between BAME and White British individuals (0.09, SE = 0.36).

Column (2) reveals that similar findings are obtained when adjusting for demographic and socioeconomic characteristics, suggesting that the larger deterioration in mental health among

**Table 4. OLS regressions of changes in mental distress in the UKHLS from Wave 9 (2017-2019) to April 2020.**

| | Difference | | Standardised difference | | Difference | |
|---|---|---|---|---|---|---|
| | GHQ-12 score | | GHQ-12 score | | GHQ "caseness" score | |
| | (1) | (2) | (3) | (4) | (5) | (6) |
| Female ($\beta$) | 0.995*** | 1.035*** | 0.166*** | 0.173*** | 0.092*** | 0.100*** |
| | (0.130) | (0.143) | (0.022) | (0.024) | (0.012) | (0.013) |
| BAME ($\gamma$) | 0.843*** | 0.921** | 0.141*** | 0.154** | 0.053** | 0.064** |
| | (0.314) | (0.360) | (0.052) | (0.060) | (0.027) | (0.031) |
| Female × BAME ($\delta$) | -0.754 | -0.820 | -0.126 | -0.137 | -0.049 | -0.062 |
| | (0.475) | (0.500) | (0.079) | (0.084) | (0.040) | (0.043) |
| Controls | NO | YES | NO | YES | NO | YES |
| Female differential change ($\gamma + \delta$) | 0.089 | 0.075 | 0.015 | 0.017 | 0.004 | 0.002 |
| between BAME and White British | (0.356) | (0.392) | (0.059) | (0.062) | (0.029) | (0.032) |
| Observations | 12,516 | 10,920 | 12,516 | 10,920 | 12,516 | 10,920 |
| R-squared | 0.007 | 0.029 | 0.007 | 0.029 | 0.007 | 0.024 |

Authors' elaboration. Observations are weighted using the survey sample weights (see S1 and S2 Appendices). Robust standard errors in parentheses.

* $p < 0.10$,

** $p < 0.05$,

*** $p < 0.01$.

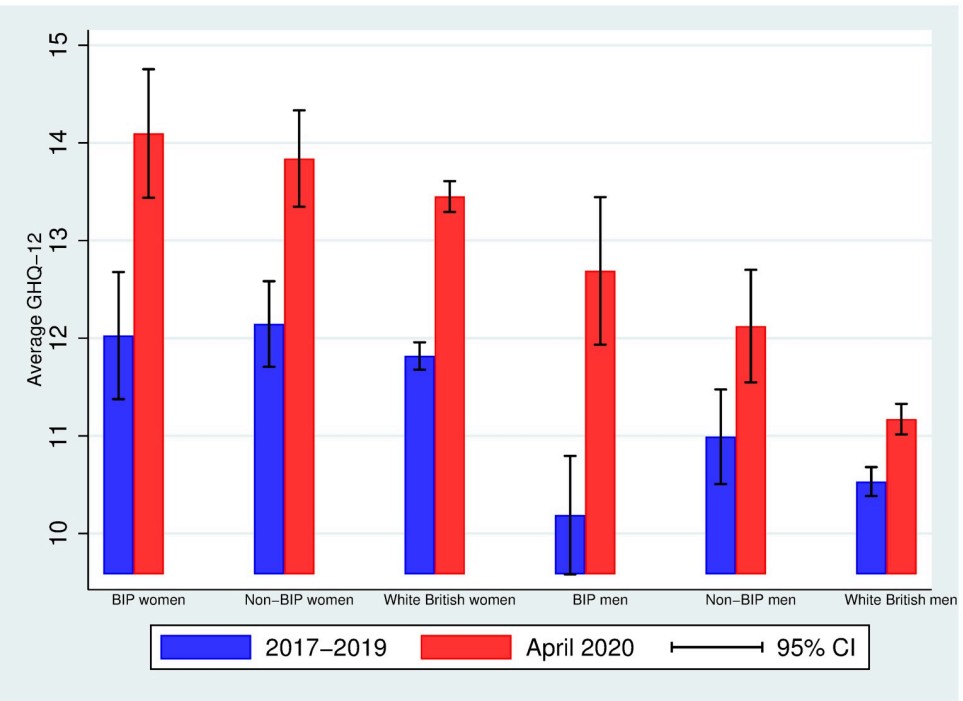

**Fig 3. Average GHQ-12 in 2017-2019 and April 2020 among BIP, non-BIP and White British individuals by gender.** Authors' elaboration using UKHLS data: wave 9 (2017-2019) and April 2020 COVID-19 wave. Observations are weighted using the survey sample weights (see S2 Appendix). 95% CI: 95% approximate confidence intervals (point estimate ± 1.96 times the standard error of the point estimate).

women and BAME men with respect to White British men cannot be accounted for by differences in: age, household size, living in a couple, qualifications, employment status, personal income, and health conditions. The set of control variables also includes place of residence (location) indicators (England (London excluded), Wales, Northern Ireland, Scotland), and interview fixed effects (month of interview fixed effects and a face-to-face indicator). The existing differences along these dimensions cannot explain the differential deterioration in mental health between 2017-2019 and April 2020 by ethnicity and gender.

Columns (3) and (4) report unadjusted and adjusted standardised differences by gender and ethnicity. As we can see in column (4), among White British individuals, women's increase in mental distress has been about 0.17 standard deviations (SE = 0.02) higher than that of men. Among men, the average deterioration in mental health among BAME has been about 0.15 standard deviations (SE = 0.06) higher than that of White British men. As previously discussed, there is no evidence of a differential increase in mental distress between BAME and White British individuals among women (0.017, SE = 0.062).

Finally, in columns (5) and (6), we investigate the change in the GHQ "caseness" score. Column (6) shows that British White women have experienced a higher increase than British White men –10 percentage points higher (SE = 1.3 percentage points, pp)– in the risk of presenting mental health problems, while BAME men have experienced a higher increase than White British men –6.4 pp higher (SE = 3.1 pp). We do not find evidence of a differential increase in the likelihood of presenting mental health problems between BAME and White British individuals among women (0.2 pp, SE = 3.2 pp).

S1 Table reports the coefficients for the control variables.

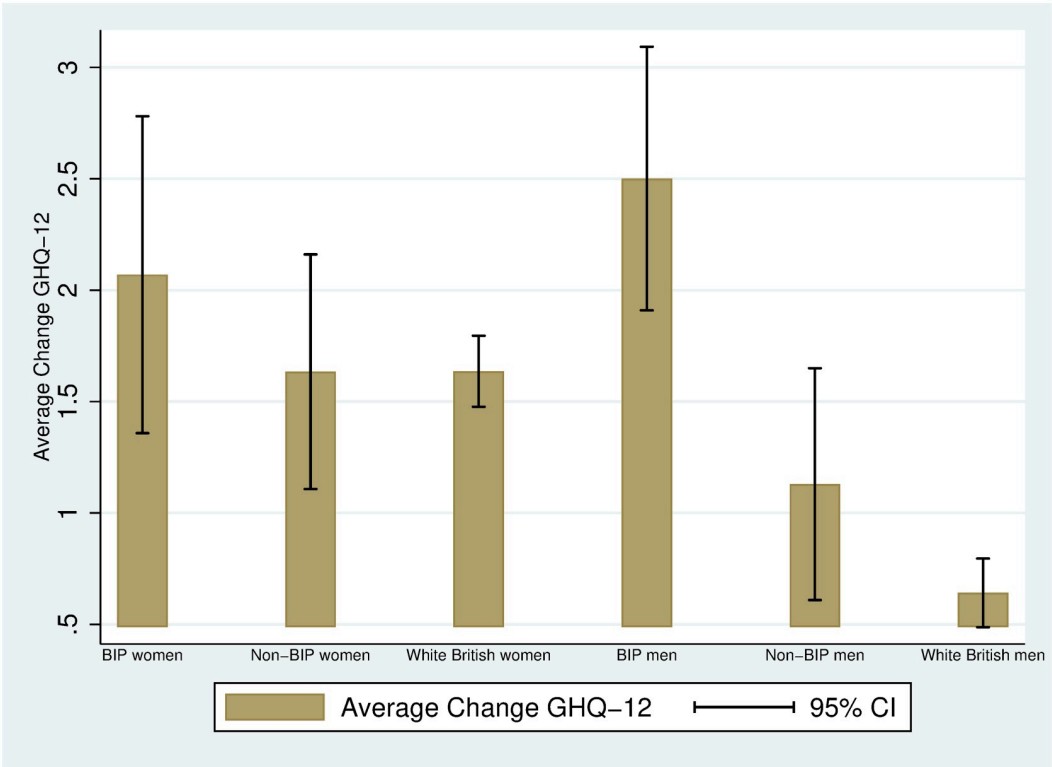

**Fig 4. Average change in GHQ-12 between 2017-2019 and April 2020 among BIP, non-BIP and White British and White British individuals by gender.** Authors' elaboration using UKHLS data: wave 9 (2017-2019) and April 2020 COVID-19 wave. Observations are weighted using the survey sample weights (see S2 Appendix). 95% CI: 95% approximate confidence intervals (point estimate ± 1.96 times the standard error of the point estimate).

### Bangladeshi, Indian and Pakistani (BIP), non-BIP vs. White British

We proceed as in the previous subsection, but focusing now on the average differences between Bangladeshi, Indian and Pakistani (BIP), BAME other than BIP (non-BIP) and White British individuals. Fig 3 displays the average mental distress (on a scale from 0-36) in 2017-2019 and April 2020 by ethnicity (BIP vs. Non-BIP vs. White British) and gender (women vs. men). As in Fig 2, women report a higher level of mental distress than men do and all groups experience an increase in mental distress from 2017-2019 to April 2020. Interestingly, the largest increase is found among BIP men: this was the group with the lowest average mental distress in 2017-2019 (10.19, 95% CI: 9.38, 10.99) and a much larger mental distress in April 2020 (12.69, 95% CI: 11.67, 13.71). Fig 4 allows us visualizing the increases in mental distress by ethnicity and gender: BIP men have experienced the highest increase in mental distress (2.5 units), and this increase is statistically different from the one experienced by both non-BIP men (1.1 units) and White British men (0.6 units), as judged by the 95% confidence intervals.

As in the case of Figs 1–4 display raw (unadjusted) differences. Table 5 displays the averages of some demographic and socioeconomic variables for men (panel A) and women (panel B) by ethnicity: BIP, non-BIP and White British. The BIP subsample is younger than the non-BIP subsample (4 years younger among men [p-value = 0.000], 8.4 years younger among women [p-value = 0.000]), and the non-BIP subsample is younger than the White British subsample (6.2 years younger among men [p-value = 0.000], 4.2 years younger among women [p-value = 0.000]). The largest average household size is found among BIP individuals. While

**Table 5. Average demographic and socioeconomic characteristics: BIP, non-BIP and White British.**

| | Age | Household size | London | BA or higher | Self-employed | Family care | Personal income | Health conditions |
|---|---|---|---|---|---|---|---|---|
| **Panel A. Men** | | | | | | | | |
| White British | 54.56 | 1.77 | 0.084 | 0.356 | 0.105 | 0.005 | 2.17 | 0.521 |
| | N = 4,541 | N = 4,541 | N = 4,539 | N = 3,991 | N = 4,525 | N = 4,525 | N = 4,541 | N = 4,541 |
| Non-BIP | 48.32 | 2.10 | 0.321 | 0.550 | 0.137 | 0.0007 | 2.24 | 0.467 |
| | N = 445 | N = 445 | N = 444 | N = 372 | N = 442 | N = 442 | N = 445 | N = 445 |
| BIP | 44.34 | 3.00 | 0.345 | 0.496 | 0.094 | 0.00 | 2.02 | 0.371 |
| | N = 267 | N = 267 | N = 267 | N = 251 | N = 267 | N = 267 | N = 267 | N = 267 |
| BIP – White British | -10.22*** | 1.23** | 0.261*** | 0.140*** | -0.011 | -0.005*** | -0.15 | -0.150*** |
| | [0.000] | [0.000] | [0.000] | [0.002] | [0.610] | [0.000] | [0.281] | [0.000] |
| BIP – Non-BIP | -3.98*** | 0.90*** | 0.024 | -0.054 | -0.043 | -0.0007 | -0.22 | -0.096* |
| | [0.009] | [0.000] | [0.637] | [0.330] | [0.174] | [0.318] | [0.190] | [0.061] |
| Non-BIP – White British | -6.24*** | 0.33** | 0.237*** | 0.194*** | 0.032 | -0.0043*** | 0.07 | -0.054*** |
| | [0.000] | [0.001] | [0.000] | [0.000] | [0.172] | [0.001] | [0.495] | [0.110] |
| **Panel B. Women** | | | | | | | | |
| White British | 51.76 | 1.83 | 0.087 | 0.317 | 0.064 | 0.046 | 1.45 | 0.514 |
| | N = 6,196 | N = 6,196 | N = 6,195 | N = 5,440 | N = 6,143 | N = 6,143 | N = 6,196 | N = 6,196 |
| Non-BIP | 47.54 | 1.86 | 0.290 | 0.489 | 0.102 | 0.070 | 1.61 | 0.419 |
| | N = 708 | N = 708 | N = 708 | N = 612 | N = 703 | N = 703 | N = 708 | N = 708 |
| BIP | 39.14 | 3.04 | 0.405 | 0.375 | 0.076 | 0.142 | 1.35 | 0.360 |
| | N = 359 | N = 359 | N = 359 | N = 331 | N = 357 | N = 357 | N = 359 | N = 359 |
| BIP – White British | -12.62*** | 1.21** | 0.318*** | 0.058 | 0.012 | 0.097*** | 0.097 | -0.155*** |
| | [0.000] | [0.000] | [0.000] | [0.132] | [0.648] | [0.000] | [0.480] | [0.000] |
| BIP – Non-BIP | -8.40*** | 1.18*** | 0.115** | -0.114** | -0.026 | 0.073*** | -0.258 | -0.060 |
| | [0.000] | [0.000] | [0.013] | [0.015] | [0.416] | [0.008] | [0.101] | [0.193] |
| Non-BIP – White British | -4.22*** | 0.03 | 0.203*** | 0.172*** | 0.038** | 0.024* | 0.161** | -0.096*** |
| | [0.000] | [0.622] | [0.000] | [0.000] | [0.026] | [0.074] | [0.049] | [0.000] |

Age (years), household size (number of people), London (0-1), BA or higher (0-1), Self-employed (0-1), Family care (0-1), Personal income (in £1,000), and Health conditions (0-1). Statistics are weighted using the survey sample weights (see S1 and S2 Appendices). p-values in brackets.

* $p < 0.10$,

** $p < 0.05$,

*** $p < 0.01$.

among men the difference in the fractions of BIP and non-BIP individuals living in London is small and not statistically significant (0.35 vs. 0.32, p-value = 0.637), BIP women are more likely to live in London than non-BIP women (0.41 vs. 0.29, p-value = 0.013). Moreover, BIP individuals tend to be less qualified than non-BIP individuals: BIP women are 11 pp less likely to hold a BA (or higher) than non-BIP women (0.38 vs. 0.49, p-value = 0.015]; among men, the average gap is 5 pp, albeit not statistically significant (0.50 vs. 0.55, p-value = 0.330). When looking at employment and personal income, among women, non-BIP individuals are 4 pp more likely to be self-employed than White British (10% vs. 6%, p-value = 0.026), and they also have, on average, a higher personal monthly income (£1,610 vs. £1,450, p-value = 0.049). We can also observe that BIP women are about 10 pp more likely to be family carers than White British women (0.14 vs. 0.046, p-value = 0.000), and about 7 pp more likely than non-BIP women (0.14 vs. 0.07, p-value = 0.008). Finally, regarding health conditions, both BIP and non-BIP individuals are less likely to report having at least one health condition than White British individuals.

**Table 6. OLS regressions of changes in mental distress in the UKHLS from Wave 9 (2017-2019) to April 2020.**

| | Difference | | Standardised difference | | Difference | |
|---|---|---|---|---|---|---|
| | GHQ-12 score | | GHQ-12 score | | GHQ "caseness" score | |
| | (1) | (2) | (3) | (4) | (5) | (6) |
| Female | 0.995*** | 1.035*** | 0.166*** | 0.173*** | 0.092*** | 0.100*** |
| | (0.130) | (0.143) | (0.022) | (0.024) | (0.012) | (0.013) |
| BIP | 1.859*** | 1.547*** | 0.310*** | 0.258*** | 0.100*** | 0.087** |
| | (0.393) | (0.425) | (0.066) | (0.071) | (0.038) | (0.040) |
| Non-BIP | 0.488 | 0.677 | 0.081 | 0.113 | 0.037 | 0.056 |
| | (0.390) | (0.452) | (0.065) | (0.076) | (0.033) | (0.038) |
| Female × BIP | -1.426** | -1.496** | -0.238** | -0.250** | -0.027 | -0.028 |
| | (0.624) | (0.657) | (0.104) | (0.110) | (0.061) | (0.062) |
| Female × Non-BIP | -0.490 | -0.564 | -0.082 | -0.094 | -0.052 | -0.068 |
| | (0.578) | (0.619) | (0.097) | (0.103) | (0.048) | (0.052) |
| Controls | NO | YES | NO | YES | NO | YES |
| Female differential change | 0.433 | 0.050 | 0.072 | 0.008 | 0.073 | 0.059 |
| between BIP and White British | (0.485) | (0.537) | (0.081) | (0.090) | (0.047) | (0.050) |
| Observations | 12,516 | 10,920 | 12,516 | 10,920 | 12,516 | 10,920 |
| R-squared | 0.007 | 0.029 | 0.007 | 0.029 | 0.007 | 0.024 |

Observations are weighted using the survey sample weights (see S1 and S2 Appendices). Robust standard errors in parentheses.

* $p < 0.10$,

** $p < 0.05$,

*** $p < 0.01$.

The differences reported in Table 5 highlight the importance of investigating whether the ethnic-gender specific changes documented in Fig 4 can be accounted for by demographic and socioeconomic characteristics. Column (1) in Table 6 replicates the graphical findings in Fig 4: BIP men have experienced a higher average increase in mental distress (1.9 units more, SE = 0.39) than White British men. The differential increase in mental distress between BIP and non-BIP men is statistically significant (p-value = 0.011, not reported in the table), and we fail to find evidence of a differential increase in mental distress between BIP and White British individuals among women (0.433, p-value = 0.485). In column (2) we add control variables: the findings reported in column (1) cannot be accounted for by differences in age, household size, living in couple, qualifications, employment status, personal income, and health conditions. However, we cannot reject the equality of the average changes in mental health between BIP and non-BIP men (p-value = 0.137).

Columns (3) and (4) report unadjusted and adjusted standardised differences by gender and ethnicity. Among men, the unadjusted average deterioration in BIP's mental health has been about 0.31 standard deviations (SE = 0.066) higher than among White British individuals, and the adjusted one has been about 0.26 standard deviations (SE = 0.071) higher.

Finally, in columns (5) and (6), we find that BIP men have experienced a higher increase than British White men –10 pp higher, SE = 3.8 pp (without controls), 8.7 pp higher, SE = 4 pp (without controls)– in the risk of presenting mental health problems. No differential change in the likelihood of having mental health problems is found between BIP and White British among women: 0.079 (SE = 0.046) without controls, 0.059 (SE = 0.050) with controls.

S2 Table reports the coefficients for the control variables. S1 File contains the files to replicate the Figures and Tables of this study.

## Discussion

The observed differences in the increase in mental distress by ethnicity and gender cannot be explained by existing differences across individuals in demographic or socioeconomic characteristics that are accounted for in the long regressions. What can then explain the ethnic-gender specific increases in mental distress between 2017-2019 and April 2020? One possibility is that individuals' mental wellbeing during the pandemic is not only affected by health concerns and financial insecurity, but also by strict physical distancing measures, such as lockdowns [26]. In the UK, the first lockdown began on the 23rd of March 2020, one month before the follow-up interview, and lockdowns are likely to have an impact on social isolation and mental health [26, 32].

A recent briefing note has unveiled that the reduction in mental wellbeing among Pakistani and Bangladeshi men with respect to White British men is less attenuated among those Pakistanis and Bangladeshis who live in areas with relatively high concentrations of own ethnic group residents [25]. Moreover, while all ethnic groups report lower levels of interpersonal contact within the neighbourhood than before the pandemic, these reductions are largest among minority ethnic groups, including Pakistanis and Bangladeshis [25]. These preliminary findings seem to be consistent with the impact of the lockdown and social distancing requirements on mental health being worse among minority ethnic groups.

Our study has two main advantages. First, we compare the same individuals before the pandemic and during the pandemic, and so we are capturing genuine average changes in mental wellbeing (e.g. no compositional bias due to comparing different groups of individuals over time). Second, questions on mental health are asked contemporaneously and not retrospectively, and so our estimates are not subject to recall bias.

Our study has three key limitations. First, the samples for different minority ethnic groups are small. This implies that our estimates are sometimes noisy, and we are also forced to investigate differences between two (BAME and White British), or three groups (BIP, non-BIP and White British). Second, our findings focus on the increase in mental distress one month into the pandemic in the UK (and one month after the UK lockdown). Whether the increase in mental distress is persistent or not, and whether such persistence varies by ethnicity and gender, is an open question. Recent research using US data shows evidence of resilience, which varies by race/ethnicity: all ethnic groups appear to go back to the initial mental health level, except for other race/ethnicity (6% of the sample) [17]. Third, while the GHQ-12 has been extensively validated in general and clinical populations worldwide, it has some well-known limitations, including low predictive value [33]. However, as long as the limitations of the GHQ-12 in measuring mental health are similar across groups defined by their ethnicity and gender, we are not concerned about obtaining a biased image of the differential increase in mental distress by ethnicity and gender between 2017-2019 and April 2020.

The second issue, the persistence or not of the deterioration in mental wellbeing and whether it varies by ethnicity and gender, can be investigated in the future using additional COVID-19 waves of the UKHLS as long as attrition is not affected by ethnicity. The first issue is much more complex. While (non-representative) surveys can be launched on online platforms such as Prolific, which allows researchers to select participants based on existing characteristics such as ethnicity, a key limitation is that participants in Prolific (or other platforms) may be different from the underlying population of interest.

We hope that our analysis and findings will emphasize the need of collecting much larger samples of minority ethnic groups so that properly-powered statistical analyses can be carried out. The same Understanding Society dataset, along other survey and administrative datasets, has its own specific ethnic minority sample (the Ethnic Minority Boost sample), which was

designed to provide at least 1,000 adult interviews from five ethnic minority groups (Indians, Pakistani, Bangladeshi, Caribbean, and African) [34].

We call for additional research on the potential differential effects of the COVID-19 pandemic by ethnicity, and urge both policy makers and researchers to allocate resources to collect larger sample sizes of minority ethnic groups. Future collection data efforts along this line will be important to investigate the potential consequences of the pandemic on both health and economic outcomes, the latter being also affected by the former via the link between wellbeing and productivity [35].

## Supporting information

**S1 Appendix. GHQ-12 questionnaire.**
(DOCX)

**S2 Appendix. Definition of variables.**
(DOCX)

**S1 Table. Table 4 with reported coefficients on control variables.**
(DOCX)

**S2 Table. Table 6 with reported coefficients on control variables.**
(DOCX)

**S1 File. Files to replicate the Tables and Figures in this article.**
(ZIP)

## Acknowledgments

This paper uses data from Understanding Society (Wave 9 and Wave April COVID-19 Study). Understanding Society is an initiative funded by the Economic and Social Research Council and various Government Departments, with scientific leadership by the Institute for Social and Economic Research, University of Essex, and survey delivery by NatCen Social Research and Kantar Public. The research data are distributed by the UK Data Service.

We thank two anonymous reviewers, Brenda Gannon, Sonia Oreffice, and seminar participants from the University of Exeter, the University of Liverpool and the University of Queensland for their helpful comments and suggestions. Any errors in this article are the sole responsibility of its authors.

## Author Contributions

**Conceptualization:** Eugenio Proto, Climent Quintana-Domeque.

**Data curation:** Eugenio Proto, Climent Quintana-Domeque.

**Formal analysis:** Eugenio Proto, Climent Quintana-Domeque.

**Investigation:** Eugenio Proto, Climent Quintana-Domeque.

**Methodology:** Eugenio Proto, Climent Quintana-Domeque.

**Writing – original draft:** Eugenio Proto, Climent Quintana-Domeque.

**Writing – review & editing:** Eugenio Proto, Climent Quintana-Domeque.

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
