## [Decision Letter · Decision Letter 0]

3 Sep 2020

PONE-D-20-24507

COVID-19 and Mental Health Deterioration among BAME groups in the UK

PLOS ONE

Dear Dr. Quintana-Domeque,

Thank you for submitting your manuscript to PLOS ONE. After careful consideration, we feel that it has merit but does not fully meet PLOS ONE’s publication criteria as it currently stands. Therefore, we invite you to submit a revised version of the manuscript that addresses the points raised during the review process.

Both reviewers like the manuscript and recommend a major revision. Their comments seem to be straightforward, and we would like for you to address them. The paper needs a better discussion on why BAME individuals in the U.K. are disproportionally affected by COVID-19. What is driving this heterogeneity?  

We look forward to receiving your revised manuscript.

Kind regards,

Gabriel A. Picone

Academic Editor

PLOS ONE

Journal Requirements:

Reviewers' comments:

Reviewer's Responses to Questions

**Comments to the Author**

1. Is the manuscript technically sound, and do the data support the conclusions?

Reviewer #1: Yes

Reviewer #2: Yes

2. Has the statistical analysis been performed appropriately and rigorously? 

Reviewer #1: Yes

Reviewer #2: Yes

3. Have the authors made all data underlying the findings in their manuscript fully available?

Reviewer #1: Yes

Reviewer #2: Yes

4. Is the manuscript presented in an intelligible fashion and written in standard English?

Reviewer #1: Yes

Reviewer #2: Yes

5. Review Comments to the Author

Reviewer #1: Summary:

This study uses data from the UK Household Longitudinal Study (UK HLS) to quantify pre vs post COVID-19 changes in mental distress by ethnicity and gender. The authors extend previous work using the same data to study the interaction between ethnicity and gender. They find that among men, Black, Asian, and Minority Ethnic (BAME) individuals experience a larger deterioration of mental health compared to British White individuals but there are no statistically significant differences by ethnicity among women. Among BAME individuals, Bangladeshi, Indian and Pakistani (BIP) individuals exhibit statistically significant larger declines in mental health compared to British White, while these differences are not statistically significant for other groups.

Comments:

1. Although previous literature has examined differences in health by ethnicity and gender, the focus of this study on the interaction between ethnicity and gender is interesting and very important to our understanding of the effects of COVID-19.

2. In the introduction, I would like to a more detailed discussion of existing literature and the British context that is relevant to this research question. The current draft provides very little motivation for the research question. The authors mention that there are concerns that UK’s minority ethnic groups are being disproportionately affected but they do not discuss why there is this concern. What do we know about the rates of exposure to infection or other risks among BAME individuals compared to British Whites? What are the underlying mechanisms that may be driving these differences? How is this related to mental health? Do we expect mental health to be worse among certain groups because of a higher exposure to infection or does the impact of infection on mental health differ by ethnicity? Similarly, why should we expect gender differences to vary across ethnic groups? Why should we expect differences across subgroups of BAME individuals (BIP, Asian, Arab, etc.)? What is the motivation for the specific research questions explored by this study?

3. Some of the discussion in the introduction related to the limitations of this study and the call for new data collection might be more appropriate in the discussion or conclusion section. The authors should consider moving this discussion to the conclusion section and instead adding more discussion of the motivation for the research question.

4. For readers who may not be familiar with the situation in the UK, it would be helpful to add some information on COVID-19 infection and mortality rates in the UK during the time of the survey (April 2020).

5. The finding that BIP individuals are mainly driving the effects for the BAME group is likely due to BIP being the largest subgroup among BAME individuals and providing enough sample size to estimate statistically significant results. All other BAME groups have substantially smaller sample sizes. In some cases, the coefficients for other BAME groups (e.g. Black) are similar to the coefficients estimated for BIP individuals. Given this, one cannot rule out the hypothesis that all BAME subgroups experienced similar declines in mental health. I think it would be helpful if the authors could do a power calculation to determine whether they have sufficient power to estimate statistically significant results for other BAME groups such as Chinese, Arabs, Blacks etc. I would also like to see the results from a regression that categorizes the BAME group in BIP and non-BIP individuals (reference category = British White). Such a regression is likely to have more statistical power to test whether mental health declines differ among BIP vs non-BIP groups.

6. What is the reason to present specification with different covariates? What do we learn from comparing the estimates from different column in a table? Which one of these is the preferred specification? Usually, such regressions can provide some information about mechanisms. However, the current draft does include any discussion of mechanisms or of how specific covariates modify the main estimates of interest. I think this should be discussed in further detail.

7. Some of the findings related to covariates are surprising. For example, why do more educated persons experience a higher increase in mental health (Table 7)? Is this due to spurious correlation? It is also surprising that factors like income and COVID risk have no impact on mental health. It would be helpful to add more discussion of these surprising results. I wonder if the lack of statistically significant estimates for key variables may also be due to low power, multicollinearity or overfitting. The authors should consider using a variable selection process such as forward or backward stepwise selection to determine the appropriate specification.

8. What is the reason or motivation for restricting the analysis to the working population? What additional information does this analysis add that we cannot get from the analysis of the general population?

Reviewer #2: The authors address an important question relating to the potential impact of COVID-19 on BAME groups using a suitable nationally representative longitudinal dataset. The study also provides a replication of key results from an initial analyses of the UKHLS COVID-19 survey.

Specific comments:

Abstract:

Please remove the phrase 'new facts' given the study reliance on a single sample with limited numbers of BAME participants and reliance on interaction effects (associated with reliability and statistical power issues). Same applies to reference to 'facts' in the introduction and other sections.

Introduction - other research that can speak to this research question should be included:

https://www.cdc.gov/mmwr/volumes/69/wr/mm6932a1.htm?s_cid=mm6932a1_w

https://psyarxiv.com/79f5v/

A more extensive account of the potential reasons why BAME groups may experience a different impact of the pandemic than other groups should be included. For instance, by providing reference to research examining race/ethnicity income and employment trends during the pandemic, wealth as a buffer, and by examining race/ethnicity and health effects (e.g. https://www.nature.com/articles/s41586-020-2521-4).

A focus of the results is on the working population - this should be justified in the introduction.

Reference should also be made to research examining the impact of traumatic experiencing by race/ethnicity (e.g.https://www.nejm.org/doi/full/10.1056/nejm200111153452024)

Given a study using the same dataset found no difference between White/non-White participants in changes in mental health (Daly et al.) the rationale for the current study is unclear. Were there specific subgroups that may be expected to show differences? Is there prior evidence to support this? No rationale for the expected heterogeneity is provided.

Discussion of study results would be better placed in the discussion section after the study method and analytical strategy have been outlined.

Method:

Extensive work has been invested in producing sampling weights for the UKHLS COVID-19 survey to address the complex survey design and issues relating to differential attrition (see https://www.iser.essex.ac.uk/research/publications/working-papers/understanding-society/2020-09). It is not clear why these weights were not used as without them the estimates are not representative of the UK. There are options to do this that are compatible with the analytical strategy used (e.g. using areg with pweights and the absorb option for fixed effects analyses?). The user guide for the UKHLS is clear on the point that use of weights should be the default approach and this needs to be addressed in the methods section "Weights are provided with these data to facilitate population inferences. If you undertake an unweighted analysis of the data, you should be clear on the assumptions that justify an unweighted analysis.".

Results:

The phrase 'non-statistically significant lower increase' should be removed as the phrase can be interpreted as implying a group difference where none has been found to exist. The below are non-significant differences in changes in mental distress between the groups mentioned and should be described as such.

Relevant sentences are "b) among females, BAME individuals experience a nonstatistically significant lower increase (-0.047 SD, 95% CI: [-0.116,0.023])"

and "(d) among BAME individuals, females experience a non-statistically significant lower increase (-0.031 SD, 95% CI: [-0.125,0.063]) in mental distress compared to men."

and "BIP individuals experience a non-statistically significant lower increase (-0.03 SD, 95% CI: [-0.164,0.105]) in mental distress compared to British White individuals;"

and "(d) among BIP individuals, females experience a non-statistically significant lower increase (-0.127 SD, 95% CI: [-0.300,0.047]) in mental distress compared to men."

and "(b) among females, BAME individuals experience a non-statistically significant lower increase (-0.038 SD, 95% CI: [- 0.144, 0.067]) in mental distress compared to British White individuals;"

and "(d) among BAME individuals, females experience a non-statistically significant lower increase (-0.054 SD, 95% CI: [-0.185, 0.077]) in mental distress compared to men."

and "(d) among BAME individuals, females experience a non-statistically significant lower increase (-2.6

pp, 95% CI: [-9.6,4.3) in mental health problems compared to men."

Discussion:

This section is currently extremely limited and requires expansion to address a range of points, including:

Potential reasons of the study findings (beyond the single reference to key worker status).

Links to prior research (e.g. research examining race/ethnicity differences in stress reactions to trauma or adversity).

Limitations of the current study, including the small BAME sample and even further reduced subsamples. The need for more extensive follow-up. For instance, the preprint above (https://psyarxiv.com/79f5v/) shows strong evidence of adaptation to the pandemic which may apply here also (though also the 'other ethnicity' group (see Table 3) was the only group remaining above baseline distress levels by July, 2020). The need for more extensive mental health assessments and so on.

Recommendations for future research. The abstract mentions collecting larger ethnic minority samples but this is not discussed extensively (e.g. by reference to oversampling).

6. PLOS authors have the option to publish the peer review history of their article (what does this mean?). If published, this will include your full peer review and any attached files.

Reviewer #1: No

Reviewer #2: No

---

## [Author Response · Author response to Decision Letter 0]

26 Nov 2020

Answers to referees

Dear reviewers,

Many thanks for your comments. 

We have substantially rewritten our paper following your comments and suggestions. 

Our answers to your comments are provided below (in red and italics).

We hope you find our substantially revised version clearer and more informative.

Best wishes,

Climent Quintana-Domeque & Eugenio Proto

Reviewer #1: Summary:

This study uses data from the UK Household Longitudinal Study (UK HLS) to quantify pre vs post COVID-19 changes in mental distress by ethnicity and gender. The authors extend previous work using the same data to study the interaction between ethnicity and gender. They find that among men, Black, Asian, and Minority Ethnic (BAME) individuals experience a larger deterioration of mental health compared to British White individuals but there are no statistically significant differences by ethnicity among women. Among BAME individuals, Bangladeshi, Indian and Pakistani (BIP) individuals exhibit statistically significant larger declines in mental health compared to British White, while these differences are not statistically significant for other groups.

Comments:

1. Although previous literature has examined differences in health by ethnicity and gender, the focus of this study on the interaction between ethnicity and gender is interesting and very important to our understanding of the effects of COVID-19.

==> Answer: Thank you.

2. In the introduction, I would like to a more detailed discussion of existing literature and the British context that is relevant to this research question. The current draft provides very little motivation for the research question. The authors mention that there are concerns that UK’s minority ethnic groups are being disproportionately affected but they do not discuss why there is this concern. What do we know about the rates of exposure to infection or other risks among BAME individuals compared to British Whites? What are the underlying mechanisms that may be driving these differences? How is this related to mental health? Do we expect mental health to be worse among certain groups because of a higher exposure to infection or does the impact of infection on mental health differ by ethnicity? Similarly, why should we expect gender differences to vary across ethnic groups? Why should we expect differences across subgroups of BAME individuals (BIP, Asian, Arab, etc.)? What is the motivation for the specific research questions explored by this study?

==> Answer: Many thanks for your comment and suggestion. Our introduction has been substantially revised along your suggested lines:

• We discuss in more detail existing studies and reports.

• We are more specific about the British context.

• We define the COVID-19 pandemic as a traumatic event following the existing literature.

• We are explicit about existing and potential differences in health (including differential risk exposure and differential vulnerability) and economic outcomes between BAME individuals compared to White British individuals, which might be relevant in explaining differences in the effect of the pandemic on mental health. 

• Similarly, we also discuss why we might expect the differences between ethnic groups to vary by gender. 

3. Some of the discussion in the introduction related to the limitations of this study and the call for new data collection might be more appropriate in the discussion or conclusion section. The authors should consider moving this discussion to the conclusion section and instead adding more discussion of the motivation for the research question.

==> Answer: Thanks for pointing this out. We have moved the discussion on the limitations of our study and call for new data collection to the last section of the paper, and we have added more discussion on the motivation in the introduction. 

4. For readers who may not be familiar with the situation in the UK, it would be helpful to add some information on COVID-19 infection and mortality rates in the UK during the time of the survey (April 2020).

==>Answer: Thanks for this important suggestion. We have added the information regarding positive cases and deaths in the UK around the time of the follow-up survey (end of April 2020). The number of cumulative positive cases in the UK by April 30 was 177,487 and the number of deaths involving the Coronavirus in March and April was 38,156.

5. The finding that BIP individuals are mainly driving the effects for the BAME group is likely due to BIP being the largest subgroup among BAME individuals and providing enough sample size to estimate statistically significant results. All other BAME groups have substantially smaller sample sizes. In some cases, the coefficients for other BAME groups (e.g. Black) are similar to the coefficients estimated for BIP individuals. Given this, one cannot rule out the hypothesis that all BAME subgroups experienced similar declines in mental health. I think it would be helpful if the authors could do a power calculation to determine whether they have sufficient power to estimate statistically significant results for other BAME groups such as Chinese, Arabs, Blacks etc. I would also like to see the results from a regression that categorizes the BAME group in BIP and non-BIP individuals (reference category = British White). Such a regression is likely to have more statistical power to test whether mental health declines differ among BIP vs non-BIP groups.

==> Answer: This is an excellent point. 

Given the power issues, and the issues in computing ex-post power calculations [A, B], we have decided to present only two types of analyses: (1) comparing BAME vs. White British individuals, and comparing (2) BIP vs. non-BIP vs. White British individuals. 

Without controls, the increase in mental distress among BIP men relative to White British men is 0.310 standard deviations (SE=0.066), while among non-BIP men relative to White British men is 0.081 standard deviations (SE=0.065). The difference between BIP and non-BIP is statistically significant (p-value=0.011). With controls, the relative increases for BIP and non-BIP are 0.258 SD (SE=0.071) and 0.113 SD (SE=0.076), and the difference is not statistically significant (p-value=0.137).

[A] http://daniellakens.blogspot.com/2014/12/observed-power-and-what-to-do-if-your.html

[B] https://statmodeling.stat.columbia.edu/2018/09/24/dont-calculate-post-hoc-power-using-observed-estimate-effect-size/

6. What is the reason to present specification with different covariates? What do we learn from comparing the estimates from different column in a table? Which one of these is the preferred specification? Usually, such regressions can provide some information about mechanisms. However, the current draft does include any discussion of mechanisms or of how specific covariates modify the main estimates of interest. I think this should be discussed in further detail.

==> Answer: This is a very good point. 

• We now only present two sets of regressions with three measures of the change in mental health (difference, standardised difference and change in GHQ-12 “caseness” score): regression without controls (which mimics the figures) and regression with the full list of controls. 

• We now clarify the purpose of comparing these two sets of regressions: “The purpose of this analysis is to show whether the ethnic-gender specific changes in mental health can be explained by differences in demographic or socioeconomic variables that can act a mediators or be affected by ethnicity characteristics. Thus, our investigation does not make causal claims. While we do not model causal chains [13], we provide a first approximation to the impact of the COVID-19 pandemic on mental health by ethnicity and gender in the UK.”

[13] Health Foundation (2020). “How to interpret research on ethnicity and COVID-19 risk and outcomes: five key questions.” 

7. Some of the findings related to covariates are surprising. For example, why do more educated persons experience a higher increase in mental health (Table 7)? Is this due to spurious correlation? It is also surprising that factors like income and COVID risk have no impact on mental health. It would be helpful to add more discussion of these surprising results. I wonder if the lack of statistically significant estimates for key variables may also be due to low power, multicollinearity or overfitting. The authors should consider using a variable selection process such as forward or backward stepwise selection to determine the appropriate specification.

==> Answer: These are very interesting points. We now report an examination of average differences in demographic and socioeconomic characteristics by ethnicity and gender. This is something we did not do in the previous version and proves to be very informative:

• Regarding the higher increase in mental distress among more educated individuals, this is consistent with previous research [16, 26]. While this finding is not central to our study (indeed, in the current version this only is statistically significant when using changes in the GHQ “caseness” score), several possibilities may explain this: first, more educated people are more engaged and interested in health information, and perhaps think more about the risks of getting COVID-19; second, high-socioeconomic groups (more educated individuals) face multiple demands on their time (job tasks, childcare and other caring responsibilities), which are specially challenging in a context of lockdown.

• The fact that income does not predict an increase in mental distress between 2017-2019 and April 2020 is consistent with the fact that in our sample the only statistically significant difference in average incomes are observed between non-BIP women and White British women (£1,610 vs. £1,450, p-value=0.049).

• The fact that having at least one health condition has no impact on mental health can somehow be related with the fact that in our sample BAME individuals are younger than White British individuals: the average gap is 7.3 years among men (p-value=0.000) and 6 among women (p-value=0.000). 

• While many of the explanatory variables are not statistically significant, this does not seem to be driven by collinearity between the independent variables, at least as judged by the variance inflation factors (VIFs). The largest VIF is 3.27 (for the age indicator 45-54) and the minimum is 1.02 (for the indicators of Wales, Northern Ireland and face-to-face interview at baseline). The mean VIF is 1.8. As a rule of thumb, a variable whose VIF values are greater than 10 may merit further investigation [C]. However, in our case, none of the variables has a value above 10, and all of them are well below 5 [D]. 

[C] Baum, C. (2006) An Introduction to Modern Econometrics Using Stata. Stata Press.

[D] Kutner M, Nachtsheim C, Neter J. (2004) Applied Linear Statistical Models. 4th. McGraw-Hill; Irwin.

8. What is the reason or motivation for restricting the analysis to the working population? What additional information does this analysis add that we cannot get from the analysis of the general population?

==> Answer: Your point is well taken. In the previous version, we were conducting two analyses: one for the general population, and one for the working population. The analysis for the working population was presented as a robustness check. However, we have decided to remove it from the revised paper.

Reviewer #2: The authors address an important question relating to the potential impact of COVID-19 on BAME groups using a suitable nationally representative longitudinal dataset. The study also provides a replication of key results from an initial analyses of the UKHLS COVID-19 survey.

Specific comments:

Abstract:

Please remove the phrase 'new facts' given the study reliance on a single sample with limited numbers of BAME participants and reliance on interaction effects (associated with reliability and statistical power issues). Same applies to reference to 'facts' in the introduction and other sections.

==> Answer: Thanks. We have rewritten the paper and removed the phrase “new facts”.

Introduction - other research that can speak to this research question should be included:

https://www.cdc.gov/mmwr/volumes/69/wr/mm6932a1.htm?s_cid=mm6932a1_w

https://psyarxiv.com/79f5v/

==> Answers: Thank you for pointing these out. These references have now been included:

[17] Daly, M., & Robinson, E. (in press). “Psychological distress and adaptation to the COVID-19 crisis in the United States.” Journal of Psychiatric Research. 

[19] Czeisler MÉ , Lane RI, Petrosky E, et al. (2020). “Mental Health, Substance Use, and Suicidal Ideation During the COVID-19 Pandemic — United States, June 24–30, 2020.” MMWR Morb Mortal Wkly Rep 2020;69:1049–1057. 

A more extensive account of the potential reasons why BAME groups may experience a different impact of the pandemic than other groups should be included. For instance, by providing reference to research examining race/ethnicity income and employment trends during the pandemic, wealth as a buffer, and by examining race/ethnicity and health effects (e.g. https://www.nature.com/articles/s41586-020-2521-4).

==> Answer: The introduction is now more explicit about existing literature and relevance of our research question. In particular, we explain some of the differences between BAME and White British individuals, and within BAME groups (and how they vary by gender) reported in previous studies or reports, regarding health and economics outcomes, and which might be relevant in explaining differences in the effect of the pandemic on mental health. 

We refer to and discuss additional work, including:

[10] Williamson, E.J., Walker, A.J., Bhaskaran, K. et al. (2020) “Factors associated with COVID-19-related death using OpenSAFELY.” Nature 584, 430–436.

A focus of the results is on the working population - this should be justified in the introduction.

Answer: Your point is well taken. In the previous version we were conducting two analyses: one for the general population, and one for the working population. The analysis for the working population was presented as a robustness check. However, we have decided to remove it from the revised paper.

Reference should also be made to research examining the impact of traumatic experiencing by race/ethnicity (e.g.https://www.nejm.org/doi/full/10.1056/nejm200111153452024)

==> Answer: We now cite recent work highlighting that the COVID-19 pandemic is a traumatic event.

[15] Ettman CK, Abdalla SM, Cohen GH, Sampson L, Vivier PM, Galea S. (2020) “Prevalence of Depression Symptoms in US Adults Before and During the COVID-19 Pandemic.” JAMA Netw Open. 2020;3(9):e2019686.

Given a study using the same dataset found no difference between White/non-White participants in changes in mental health (Daly et al.) the rationale for the current study is unclear. Were there specific subgroups that may be expected to show differences? Is there prior evidence to support this? No rationale for the expected heterogeneity is provided.

==> Answer: We are now clearer on the motivation of our study. The introduction is explicit about why we may expect differences across different ethnic groups. For instance, we now write:

“This differential mortality risk by ethnicity can be driven by a higher risk of acquiring infection (e.g., if ethnic minority individuals are more likely to be employed as “key workers”, which are subject to a higher risk of infection), a higher risk of poor outcomes once infected (e.g., if ethnic minority individuals are more likely to suffer from underlying health conditions), or both [13]. For instance, the Indian ethnic group represents 14% of doctors in England and Wales, but only 3% of the working-age population [12], and recent reports show that the average Black, African and Ethnic Minority (BAME) risk of infection is 56% higher than the White British risk for working-age people, and 69% higher for those 65 plus [14]. Bangladeshis are more than 60% more likely to have a long-term health condition compared with White British aged 60 plus [12]. However, only a small part of the excess COVID-19 mortality risk of ethnic minority groups can be explained by comorbidities, deprivations, or other factors [10].

The differential economic impact by ethnicity can be driven by unemployment, income loss, or financial insecurity. Pakistani men are 70% more likely to be self-employed than the White British majority [12], and the incomes of self-employed workers are more uncertain. In addition, men from minority ethnic groups are more likely to be affected by the shutdown [12]: Bangladeshi men are four times as likely as White British men to have jobs in shut-down industries (e.g. restaurant sector), and Pakistani men are nearly three times as likely as White British men (e.g. taxi driving sector).

The COVID-19 pandemic can be considered a traumatic event [15], which may lead to mental health deterioration for multiple reasons [15-17]. In the UK, there has been an increase in mental distress between the pre-pandemic and pandemic periods, stronger among women and younger individuals [18]. Given the differences in mortality risk and financial security across different ethnic groups, we may expect differential effects on mental health too. Indeed, in the US, racial/ethnic minorities reported having experienced disproportionately worse mental health outcomes [19]. While in the UK there were no differential changes in mental health problems between White and non-White individuals from 2017-2019 to April 2020 [16], this comparison may mask important differences across ethnic groups [12, 16]. Moreover, while a lot has been documented on gender inequality and the pandemic [18, 20-22], the potential interaction between gender and ethnicity requires further investigation, if only because of the different “gender roles” within households across different ethnic groups. For instance, 29% of Bangladeshi working-age men both work in a shut-down sector and have a partner who is not in paid work compared with only 1% of White British men [12].” 

Discussion of study results would be better placed in the discussion section after the study method and analytical strategy have been outlined.

==> Answer: We have moved this to the appropriate section.

Method:

Extensive work has been invested in producing sampling weights for the UKHLS COVID-19 survey to address the complex survey design and issues relating to differential attrition (see https://www.iser.essex.ac.uk/research/publications/working-papers/understanding-society/2020-09). It is not clear why these weights were not used as without them the estimates are not representative of the UK. There are options to do this that are compatible with the analytical strategy used (e.g. using areg with pweights and the absorb option for fixed effects analyses?). The user guide for the UKHLS is clear on the point that use of weights should be the default approach and this needs to be addressed in the methods section "Weights are provided with these data to facilitate population inferences. If you undertake an unweighted analysis of the data, you should be clear on the assumptions that justify an unweighted analysis.".

==> Answer: 

We have now used weights following https://www.iser.essex.ac.uk/research/publications/working-papers/understanding-society/2020-09 and https://www.youtube.com/watch?v=6xwrIdUmxts&feature=youtu.be

Reassuringly, the main documented differences in the change in mental wellbeing from 2017-2019 to April 2020 are robust to using weights (the revised version of the paper) or not using them (the previous version of the paper).

Results:

The phrase 'non-statistically significant lower increase' should be removed as the phrase can be interpreted as implying a group difference where none has been found to exist. The below are non-significant differences in changes in mental distress between the groups mentioned and should be described as such.

Relevant sentences are "b) among females, BAME individuals experience a nonstatistically significant lower increase (-0.047 SD, 95% CI: [-0.116,0.023])"

and "(d) among BAME individuals, females experience a non-statistically significant lower increase (-0.031 SD, 95% CI: [-0.125,0.063]) in mental distress compared to men."

and "BIP individuals experience a non-statistically significant lower increase (-0.03 SD, 95% CI: [-0.164,0.105]) in mental distress compared to British White individuals;"

and "(d) among BIP individuals, females experience a non-statistically significant lower increase (-0.127 SD, 95% CI: [-0.300,0.047]) in mental distress compared to men."

and "(b) among females, BAME individuals experience a non-statistically significant lower increase (-0.038 SD, 95% CI: [- 0.144, 0.067]) in mental distress compared to British White individuals;"

and "(d) among BAME individuals, females experience a non-statistically significant lower increase (-0.054 SD, 95% CI: [-0.185, 0.077]) in mental distress compared to men."

and "(d) among BAME individuals, females experience a non-statistically significant lower increase (-2.6

pp, 95% CI: [-9.6,4.3) in mental health problems compared to men."

==> Answer: We have removed sentences such as “non-statistically significant lower increase”.

Discussion:

This section is currently extremely limited and requires expansion to address a range of points, including:

Potential reasons of the study findings (beyond the single reference to key worker status).

Links to prior research (e.g. research examining race/ethnicity differences in stress reactions to trauma or adversity).

Limitations of the current study, including the small BAME sample and even further reduced subsamples. The need for more extensive follow-up. For instance, the preprint above (https://psyarxiv.com/79f5v/) shows strong evidence of adaptation to the pandemic which may apply here also (though also the 'other ethnicity' group (see Table 3) was the only group remaining above baseline distress levels by July, 2020). The need for more extensive mental health assessments and so on.

Recommendations for future research. The abstract mentions collecting larger ethnic minority samples but this is not discussed extensively (e.g. by reference to oversampling).

==> Answer: We have now expanded the discussion/conclusions section. 

• Potential reasons of the study findings and links to previous research:

“[…] One possibility is that individuals’ mental wellbeing during the pandemic is not only affected by health concerns and financial insecurity, but also by strict physical distancing measures, such as lockdowns [26]. In the UK, the first lockdown began on the 23rd of March 2020, one month before the follow-up interview, and lockdowns are likely to have an impact on social isolation and mental health [26, 32].

A recent briefing note has unveiled that the reduction in mental wellbeing among Pakistani and Bangladeshi men with respect to White British men is less attenuated among those Pakistanis and Bangladeshis who live in areas with relatively high concentrations of own ethnic group residents [25]. Moreover, while all ethnic groups report lower levels of interpersonal contact within the neighbourhood than before the pandemic, these reductions are largest among minority ethnic groups, including Pakistanis and Bangladeshis [25]. These preliminary findings seem to be consistent with the impact of the lockdown and social distancing requirements on mental health being worse among minority ethnic groups.”

Limitations of our study:

“Our study has three key limitations. First, the samples for different minority ethnic groups are small. This implies that our estimates are sometimes noisy, and we are also forced to investigate differences between two (BAME and White British), or three groups (BIP, non-BIP and White British). Second, our findings focus on the increase in mental distress one month into the pandemic in the UK (and one month after the UK lockdown). Whether the increase in mental distress is persistent or not, and whether such persistence varies by ethnicity and gender, is an open question. Recent research using US data shows evidence of resilience, which varies by race/ethnicity: all ethnic groups appear to go back to the initial mental health level, except for other race/ethnicity (6% of the sample) [17]. Third, while the GHQ-12 has been extensively validated in general and clinical populations worldwide, it has some well-known limitations, including low predictive value [33]. However, as long as the limitations of the GHQ-12 in measuring mental health are similar across groups defined by their ethnicity and gender, we are not concerned about obtaining a biased image of the differential increase in mental distress by ethnicity and gender between 2017-2019 and April 2020.

 The second issue, the persistence or not of the deterioration in mental wellbeing and whether it varies by ethnicity and gender, can be investigated in the future using additional COVID-19 waves of the UKHLS as long as attrition is not affected by ethnicity. The first issue is much more complex. While (non-representative) surveys can be launched on online platforms such as Prolific, which allows researchers to select participants based on existing characteristics such as ethnicity, a key limitation is that participants in Prolific (or other platforms) may be different from the underlying population of interest.”

Recommendations for future research:

“We hope that our analysis and findings will emphasize the need of collecting much larger samples of minority ethnic groups so that properly-powered statistical analyses can be carried out. The same Understanding Society dataset, along other survey and administrative datasets, has its own specific ethnic minority sample (the Ethnic Minority Boost sample), which was designed to provide at least 1,000 adult interviews from five ethnic minority groups (Indians, Pakistani, Bangladeshi, Caribbean, and African) [34]. 

We call for additional research on the potential differential effects of the COVID-19 pandemic by ethnicity, and urge both policy makers and researchers to allocate resources to collect larger sample sizes of minority ethnic groups. Future collection data efforts along this line will be important to investigate the potential consequences of the pandemic on both health and economic outcomes, the latter being also affected by the former via the link between wellbeing and productivity [35].” 

---

## [Decision Letter · Decision Letter 1]

10 Dec 2020

COVID-19 and mental health deterioration by ethnicity and gender in the UK

PONE-D-20-24507R1

Dear Dr. Quintana-Domeque,

We’re pleased to inform you that your manuscript has been judged scientifically suitable for publication and will be formally accepted for publication once it meets all outstanding technical requirements.

Kind regards,

Gabriel A. Picone

Academic Editor

PLOS ONE

Additional Editor Comments (optional):

Reviewers' comments:

Reviewer's Responses to Questions

**Comments to the Author**

1. If the authors have adequately addressed your comments raised in a previous round of review and you feel that this manuscript is now acceptable for publication, you may indicate that here to bypass the “Comments to the Author” section, enter your conflict of interest statement in the “Confidential to Editor” section, and submit your "Accept" recommendation.

Reviewer #1: All comments have been addressed

Reviewer #2: All comments have been addressed

2. Is the manuscript technically sound, and do the data support the conclusions?

Reviewer #1: Yes

Reviewer #2: Yes

3. Has the statistical analysis been performed appropriately and rigorously? 

Reviewer #1: Yes

Reviewer #2: Yes

4. Have the authors made all data underlying the findings in their manuscript fully available?

Reviewer #1: Yes

Reviewer #2: Yes

5. Is the manuscript presented in an intelligible fashion and written in standard English?

Reviewer #1: Yes

Reviewer #2: Yes

6. Review Comments to the Author

Reviewer #1: (No Response)

Reviewer #2: The authors have done an excellent job in thoroughly revising the manuscript and addressing this reviewers concerns. If possible, the one suggestion I would have is to restate the precise nature of the ethnic-gender specific increases in distress at the beginning of the discussion section.

7. PLOS authors have the option to publish the peer review history of their article (what does this mean?). If published, this will include your full peer review and any attached files.

Reviewer #1: No

Reviewer #2: No

---

## [Editor Report · Acceptance letter]

14 Dec 2020

PONE-D-20-24507R1 

COVID-19 and mental health deterioration by ethnicity and gender in the UK 

Dear Dr. Quintana-Domeque:

I'm pleased to inform you that your manuscript has been deemed suitable for publication in PLOS ONE. Congratulations! Your manuscript is now with our production department. 

Kind regards, 

on behalf of

Dr. Gabriel A. Picone 

Academic Editor

PLOS ONE